# Local Density of States Correlations in the Lévy-Rosenzweig-Porter random matrix ensemble

A.V. Lunkin[1*], K.S. Tikhonov[2]

**1** Nanocenter CENN, Jamova 39, Ljubljana, SI-1000, Slovenia
**2** Capital Fund Management, 23 rue de l'Université, 75007 Paris, France
* Aleksey.Lunkin@nanocenter.si

October 21, 2024

## Abstract

We present an analytical calculation of the local density of states correlation function $\beta(\omega)$ in the Lévy-Rosenzweig-Porter random matrix ensemble at energy scales larger than the level spacing but smaller than the bandwidth. The only relevant energy scale in this limit is the typical level width $\Gamma_0$. We show that $\beta(\omega \ll \Gamma_0) \sim W/\Gamma_0$ (here $W$ is width of the band) whereas $\beta(\omega \gg \Gamma_0) \sim (W/\Gamma_0)(\omega/\Gamma_0)^{-\mu}$ where $\mu$ is an index characterising the distribution of the matrix elements. We also provide an expression for the average return probability at long times: $\ln[R(t \gg \Gamma_0^{-1})] \sim -(\Gamma_0 t)^{\mu/2}$. Numerical results based on the pool method and exact diagonalization are also provided and are in agreement with the analytical theory.

# 1   Introduction

The Anderson localisation [1] is one of the milestones in the physics of disordered systems. While the non-interacting case of the localisation is well studied (see e.g. [2]), the influence of the interaction on the localisation is not yet fully understood. It has been shown that diffusion stops in interacting systems even at infinite temperature at sufficiently strong disorder [3, 4, 5]. This phenomenon is known as many-body localisation (MBL). One fruitful approach to study the MBL is to think of a Hamiltonian of MBL problem as a tight-binding model defined in the Fock space. To facilitate analytical progress, certain simplifications should be made to this model (most importantly, neglect or simplification of the correlation structure of the exponentially large number of the matrix elements of this tight-binding model) [6, 7, 8, 9, 10, 11, 12].

Recently it has been shown that the 'resonance frequencies' in MBL problems are broadly distributed [13, 14], and even demonstrate the power–law tail [15], suggesting that random matrix ensembles with power-law-distributed off-diagonal elements[16, 17, 18, 19] can be relevant for certain properties of the many-body problems. The goal of the present paper is to study one model of this class, the Lévy-Rosenzweig-Porter (LRP) model. The key feature of the LRP model in contrast to the Rosenzweig-Porter (RP) model [20], where distribution of the off-diagonal matrix elements is Gaussian, is emergence of the non-trivial distribution of the imaginary part of the local self-energy $\gamma$. This shows up, for example, in the behaviour of the return probability which, for the RP model reads $R(t) = |\langle j|e^{-iHt}|j\rangle|^2 \sim e^{-\gamma t}$ where $\gamma$ is given by the Fermi Golden Rule (FGR). In contrast, as we will show, for the LRP model, $\gamma$ becomes a random variable distributed according to $\ln\{P(\gamma \to 0)\} \sim -(\gamma/\Gamma_0)^{-\mu/(2-\mu)}$. This fact leads to the stretched exponential behaviour of the return probability: $\ln(1/R(t)) \sim (\Gamma_0 t)^{\mu/2}$, where $\Gamma_0$ is the typical level width. While this consideration gives the correct stretch exponent, its a little bit naive since FGR does not apply to this problem (the variance of the off-diagonal matrix elements is infinite!). In the present paper, we provide a complete analytical theory for the return probability and compare it to the results of numerical simulations of the same model.

The structure of the article is as follows. The section 2 is devoted to introduction of the LRP model and the 'semi-classical' approximation which is crucial in our analysis. In the Section 3, which is the most technical part of our work, we derive the distribution of the

imaginary part of the local self-energy. Using this distribution, in the Section 4 we evaluate the finite-frequency correlation function of the local density of states (LDOS) and return probability as a function of time. The Section 5 is devoted to the numerical analysis of the LRP model and comparison to analytical results. We discuss our results from the qualitative point of view in the Section 6. Finally, Appendices conclude this work providing some further technical details.

## 2 Preliminaries

### 2.1 Model

The LRP random matrix ensemble was introduced in [18]. The ensemble has the following form:

$$H = H_D + H_L. \tag{1}$$

In this equation, $H_D$ is a diagonal matrix with i.i.d. random matrix elements, drawn from the distribution $P_D(\xi)$. The particular shape of this distribution is not important for our analytical consideration. We assume that it is centred and the width of this distribution is $W$. The matrix $H_L$ is a symmetric matrix whose elements are i.i.d. random values drawn from the fat-tailed distribution $P_L^\mu(h)$. The particular form of this distribution is not important, as all properties of the model are determined by the large-argument asymptotics. This asymptotic is conventionally [18, 19] chosen as follows:

$$P_L^\mu(h) \approx \frac{\mu}{2N^\eta |h|^{\mu+1}} \quad \text{at} \quad |h| \to \infty. \tag{2}$$

For $\mu > 2$ this distribution has a finite variance and the model becomes equivalent to the RP model. The typical decay rate for the LRP model for $2 > \mu > 1$ was estimated in [19] with the result is $\text{Im}\,\Sigma_{typ} \sim N^{(1-\eta)/(\mu-1)}$. The distribution in Eq. (2) leads to a rich phase diagram in the parameter space of $(\mu, \eta)$. In addition to the localized and fully ergodic phases, a new phase emerges for $\mu > \eta > 1$. This phase is extended, but a typical state occupies only a vanishing fraction of the full Hilbert space. This phase is known as the non-ergodic extended phase [20, 19]. For our study, we find it more natural to have a finite decay rate in the limit of $N \to \infty$, so we prefer to define the model as follows [16]:

$$P_L^\mu(h) \approx \frac{h_0^\mu \mu}{2N |h|^{\mu+1}} \quad |h| \to \infty. \tag{3}$$

Formally, the two definitions are related by the replacement $h_0^\mu \leftrightarrow N^{1-\eta}$.

In what follows, we will focus our analysis on the case $2 > \mu > 1$ and, in most of the paper, on the limit of $h_0 \ll W$. In the opposite limit of $W \to 0$, the term $H_D$ is irrelevant and one arrives at the Lévy random ensemble[16]. In our case, the presence of the strong diagonal disorder ($h_0 \ll W$) makes the analytical treatment somewhat easier, since the problem becomes essentially quasi-classical. At the same time, a new energy scale $\Gamma_0 \ll W$, controlling the LDOS correlations and behaviour of the return probability, emerges in the problem.

## 2.2 Green function and cavity equation

We are interested in the distribution of the local Green function $G$ and the self energy $\Sigma$:

$$G_{ab}(z) = \langle a|(z-H)^{-1}|b\rangle \qquad \Sigma_j(z) = \frac{1}{G_{jj}(z)} - z - H_{jj}. \tag{4}$$

The main tool we are going to use is the cavity equation derived from the following recursion identity:

$$\frac{1}{G_{jj}(z)} = z - H_{jj} - \sum_{l,m\neq j} H_{jl}G_{lm}^{\hat{j}}(z)H_{mj}, \tag{5}$$

where $G_{lm}^{\hat{j}}(z)$ is a Green function for the matrix which was obtained from $H$ by crossing out $j$th row and $j$th column. In the limit $N \to \infty$ the terms with $l \neq m$ can be neglected [16, 21] (see also Appendix B). Under this assumption the recursion relation simplifies to:

$$\frac{1}{G_{jj}(z)} = z - H_{jj} - \sum_{l\neq j} |H_{jl}|^2 G_{ll}^{\hat{j}}(z) \iff \Sigma_j(z) = \sum_{l\neq j} |H_{jl}|^2 G_{ll}^{\hat{j}}(z). \tag{6}$$

This relation helps us to describe the distribution of the self-energy under the assumption that the statistical properties of matrices of size $N$ and $N+1$ are indistinguishable in the limit $N \to \infty$.

## 2.3 Local density of states

The LDOS is defined as:

$$\rho_j(\epsilon) = -\frac{1}{\pi} \lim_{\delta \to +0} \operatorname{Im} G_{jj}(\epsilon + i\delta). \tag{7}$$

Note that the limits $N \to \infty$ and $\delta \to 0$ do not commute and we consider the limit $N \to \infty$ to be taken first (in particular, we aim to describe the physical properties of the model on the scales larger than the level spacing). In what follows, we will omit the position index of the local quantities (such as in $\rho_j(\epsilon)$) as long as it is not leading to confusion.

The behaviour of the average LDOS simplifies in the limit of $h_0 \ll W$:

$$\langle \rho(\epsilon) \rangle = -\frac{1}{\pi} \int d\xi P_D(\xi) \langle \operatorname{Im}\left[\frac{1}{\epsilon + i\delta - \xi - \Sigma(\epsilon + i\delta)}\right] \rangle, \tag{8}$$

where we introduced $\langle \ldots \rangle$ for ensemble averaging. As we will show, $\operatorname{Re}\Sigma_j \sim h_0^2/W \ll W$ (see also Ref. [22]). This means that the integral is coming from $\xi \approx \epsilon$. This 'semi-classical approximation' is crucial for our analysis and will be used in what follows. In this approximation we find $\langle \rho_j(\epsilon) \rangle = P_D(\epsilon)$. In other words, in the limit $h_0 \ll W$ the re-normalisation of the averaged spectrum due to off-diagonal matrix elements is negligible. However off-diagonal terms create LDOS correlations which we are going to study.

## 2.4 LDOS correlation function

The main object of our interest is the correlation function of the LDOS:

$$\beta(\omega, \epsilon) = \frac{\langle \rho(\epsilon + \omega/2)\rho(\epsilon - \omega/2)\rangle}{\langle \rho(\epsilon)\rangle^2}. \tag{9}$$

In particular, Fourier transform of this quantity determines the return probability as a function of time (see Sec. 4.2). In the semi-classical approximation we find:

$$\beta(\omega, \epsilon) \approx \beta_+(\omega, \epsilon) + \beta_-(\omega, \epsilon), \tag{10}$$

where

$$\beta_+(\omega, \epsilon) = \beta_-(-\omega, \epsilon) = \frac{\langle G(\epsilon + \omega/2 + i\delta) G(\epsilon - \omega/2 - i\delta) \rangle}{(2\pi P_D(\epsilon))^2}. \tag{11}$$

Since the function $\beta_+(\omega, \epsilon)$ is analytic in the upper-half plane of $\omega$, it's convenient to introduce an imaginary frequency: $i\varkappa = \omega + i\delta$, perform all calculation assuming that $\varkappa$ is a positive real quantity and at the end perform an analytical continuation back to the real frequency. We start with:

$$\beta_+(i\varkappa, \epsilon) = \int \frac{P_D(\xi) d\xi}{(2\pi P_D(\epsilon))^2} \langle \frac{1}{\epsilon + i\varkappa/2 - \xi - \Sigma_j(\epsilon + i\varkappa/2)} \frac{1}{\epsilon - i\varkappa/2 - \xi - \Sigma_j(\epsilon - i\varkappa/2)} \rangle. \tag{12}$$

In the semi-classical approximation, we have $\Sigma(\epsilon + i\varkappa/2) + \Sigma(\epsilon - i\varkappa/2) \sim h_0^2/W \ll W$ which gives:

$$\beta_+(i\varkappa, \epsilon) = \frac{P_D(\epsilon)}{(2\pi P_D(\epsilon))^2} \int d\xi \langle \frac{1}{\xi^2 + [(\gamma(\varkappa, \epsilon) + \varkappa)/2]^2} \rangle = \frac{1}{2\pi P_D(\epsilon)} \langle \frac{1}{\varkappa + \gamma(\varkappa, \epsilon)} \rangle, \tag{13}$$

where we have introduced the energy– and (imaginary) frequency-dependent level width:

$$\gamma(\varkappa, \epsilon) = i(\Sigma(\epsilon + i\varkappa/2) - \Sigma(\epsilon - i\varkappa/2)). \tag{14}$$

In this equation, $\gamma(\varkappa, \epsilon)$ is a real positive (as $\varkappa > 0$) random variable. The Eq. (13) reminds the Breit–Wigner line shape, however the distribution of $\gamma(\varkappa, \epsilon)$ depends on $\varkappa$. To complete the calculation in Eq. (14) we should find a distribution function of $\gamma$, which is the aim of the next section.

## 3 The distribution of the level widths

At this stage, we are set to compute the probability distribution of $\gamma$, defined in Eq. (14). The reader may find useful the reminder about the generalised central limit theorem and properties of the one-sided stable distributions which can be found in the Appendix A. We start with

$$P(\gamma | \varkappa, \epsilon) = \langle \delta \left[ \gamma - i(\Sigma(\epsilon + i\varkappa/2) - \Sigma(\epsilon - i\varkappa/2)) \right] \rangle. \tag{15}$$

The $\gamma$ is a positive quantity, so it is convenient to use the Laplace transform to represent its probability density:

$$P(\gamma | \varkappa, \epsilon) = \int\limits_{-i\infty}^{i\infty} \frac{d\tau}{2\pi i} \langle e^S \rangle \qquad S = \tau(\gamma - i(\Sigma(\epsilon + i\varkappa/2) - \Sigma(\epsilon - i\varkappa/2))). \tag{16}$$

Using the cavity equation Eq. (6) we can rewrite the difference of the self-energies as follows:

$$i(\Sigma(\epsilon + i\varkappa/2) - \Sigma(\epsilon - i\varkappa/2)) = i \sum_{l \neq j} H_{lj}^2 \left[ G_{ll}^{\hat{j}}(\epsilon + i\varkappa/2) - G_{ll}^{\hat{j}}(\epsilon - i\varkappa/2) \right]. \tag{17}$$

Note that $H_{lj}$ and $G_{ll}^{\hat{j}}$ are independent random variables. We can take the average over $|H_{jl}|^2$ in the above integral in the limit $N \to \infty$ (see the Appendix A.2). The resulting 'action' $S$ reads:

$$S = \tau\gamma + \frac{h_0^\mu \mu}{2N}\Gamma(-\mu/2)\sum_l \left[i\tau\left(G_{ll}^{\hat{j}}(\epsilon + i\varkappa/2) - G_{ll}^{\hat{j}}(\epsilon - i\varkappa/2)\right)\right]^{\mu/2}. \tag{18}$$

In the large-$N$ limit, the above sum is a self-averaging quantity, so we can replace the above sum by its average. Note also that the distributions of $G_{ll}^{\hat{j}}$ and $G_{ll}$ are identical in the large-$N$ limit. Finally:

$$P(\gamma|\varkappa, \epsilon) = \int\limits_{-i\infty}^{i\infty} \frac{d\tau}{2\pi i} e^S \qquad S = \tau\gamma - \tau^{\mu/2}\gamma_0(\varkappa, \epsilon)^{\mu/2}$$

$$\gamma_0(\varkappa, \epsilon)^{\mu/2} = h_0^\mu \Gamma\left(1 - \frac{\mu}{2}\right)\langle[i\left(G_{ll}(\epsilon + i\varkappa/2) - G_{ll}(\epsilon - i\varkappa/2)\right)]^{\mu/2}\rangle. \tag{19}$$

This is a probability density function of the one-sided stable distribution with the index $\mu/2$ (see Eq. (59)), i.e:

$$P(\gamma|\varkappa, \epsilon) = \frac{1}{\gamma_0(\varkappa, \epsilon)}L_{\mu/2}(\gamma/\gamma_0(\varkappa, \epsilon)), \tag{20}$$

where $\gamma_0(\varkappa, \epsilon)$ is a typical value of the level width. The similar equation can be derived for the marginal distribution of the real part of self-energy:

$$R(\epsilon, \varkappa) = \frac{1}{2}\left(\Sigma(\varepsilon + \frac{i\varkappa}{2}) + \Sigma(\varepsilon - \frac{i\varkappa}{2})\right), \tag{21}$$

which is also a stable distribution:

$$P(R|\varkappa, \epsilon) = \frac{1}{R_0(\varkappa, \epsilon)}L_{\mu/2}^{\beta(\varkappa,\epsilon)}(R/R_0(\varkappa, \epsilon)) \tag{22}$$

where $R_0(\varkappa, \epsilon)$ is a typical value of the real-part of the self-energy. The formula for $L_{\mu/2}^{\beta(\varkappa,\epsilon)}$ is given in the Appendix C. The energy scales $\gamma_0$ and $R_0$ can be found, assuming the joint probability function for real and imaginary part $P(R, \gamma|\varkappa, \epsilon)$ is known:

$$\gamma_0(\varkappa, \epsilon)^{\mu/2} = h_0^\mu\Gamma(1 - \mu/2)\int P_D(\xi)d\xi \int dRd\gamma P(R, \gamma|\varkappa, \epsilon)F_\varkappa^{(1)}(\gamma, \epsilon - R - \xi) \tag{23}$$

and

$$R_0(\varkappa, \epsilon)^{\mu/2}e^{-i\beta(\varkappa,\epsilon)} = h_0^\mu\Gamma(1 - \mu/2)\int P_D(\xi)d\xi \int dRd\gamma P(R, \gamma|\varkappa, \epsilon)F_\varkappa^{(2)}(\gamma, \epsilon - R - \xi) \tag{24}$$

where

$$F_\varkappa^{(1)}(\gamma, \epsilon) = \left(\frac{\varkappa + \gamma}{\epsilon^2 + \left(\frac{\varkappa+\gamma}{2}\right)^2}\right)^{\mu/2} \tag{25}$$

and

$$F_\varkappa^{(2)}(\gamma, \epsilon) = \left(\frac{|\epsilon|}{\epsilon^2 + \left(\frac{\varkappa+\gamma}{2}\right)^2}\right)^{\mu/2} e^{i\pi\, \text{sgn}(\epsilon)\mu/4}. \tag{26}$$

In the semi-classical approximation, a simplified self-consistency equation can be derived:

$$\gamma_0(\varkappa, \epsilon)^{\mu/2} \approx h_0^\mu \Gamma(1 - \mu/2) P_D(\epsilon) \int\limits_{-\infty}^{\infty} d\xi \int dR d\gamma P(R, \gamma | \varkappa, \epsilon) \left[ \frac{\varkappa + \gamma}{(\xi + R)^2 + \left(\frac{\varkappa + \gamma}{2}\right)^2} \right]^{\mu/2}, \quad (27)$$

where we used a definition of the level width, Eq. (14). Performing integration in $\xi$, we notice that the result depends not on the whole joint distribution but only on the marginal distribution of $\gamma$ which is known, see Eq. (20). Finally, we arrive at

$$\gamma_0(\varkappa, \epsilon)^{\mu-1} = h_0^\mu \Gamma\left(1 - \frac{\mu}{2}\right) P_D(\epsilon) 2^{\mu-1} \frac{\Gamma(\frac{\mu-1}{2})\Gamma(\frac{1}{2})}{\Gamma(\frac{\mu}{2})} \int\limits_{0}^{\infty} dr L_{\mu/2}(r) \left(\frac{\varkappa}{\gamma_0(\varkappa, \epsilon)} + r\right)^{1-\mu/2}. \quad (28)$$

This self-consistency equation for $\gamma_0$ is central to our analysis. It is instructive to split it in two equations. The first equation defines the main frequency scale in our problem $\Gamma_0(\epsilon) = \gamma_0(0, \epsilon)$:

$$\Gamma_0(\epsilon)^{\mu-1} = h_0^\mu P_D(\epsilon) 2^{\mu-1} \Gamma\left(1 - \frac{\mu}{2}\right) \frac{\Gamma(\frac{\mu-1}{2})\Gamma(\frac{1}{2})}{\Gamma(\frac{\mu}{2})} \frac{\Gamma(2 - 2/\mu)}{\Gamma(\frac{\mu}{2})}. \quad (29)$$

(the technical details of the calculation of integrals with a one-sided stable distribution can be found in the Appendix A.3). The second equation becomes a self-consistent equation for the level width measured in 'natural' units:

$$\left(\frac{\gamma_0(\varkappa, \epsilon)}{\Gamma_0(\epsilon)}\right)^{\mu-1} = \frac{\Gamma(\mu/2)}{\Gamma(2 - 2/\mu)} \int\limits_{0}^{\infty} dr L_{\mu/2}(r) \left(\frac{\varkappa}{\gamma_0(\varkappa, \epsilon)} + r\right)^{1-\mu/2}. \quad (30)$$

The Eqs. (29), (30) fully describe the distribution of level widths required for computation of the LDOS correlation function, Eq. (13). The Eq. (30) can be solved explicitly in the large-$\varkappa$ limit:

$$\left(\frac{\gamma_0(\varkappa, \epsilon)}{\Gamma_0(\epsilon)}\right)^{\mu/2} \approx \frac{\Gamma(\mu/2)}{\Gamma(2 - 2/\mu)} \left(\frac{\varkappa}{\Gamma_0(\epsilon)}\right)^{1-\mu/2} \quad \varkappa \gg \Gamma_0. \quad (31)$$

## 4 LDOS correlation function and return probability

In this Section we use the results for the level width distribution, derived above, to evaluate the LDOS correlation function, Eq. (9) and then the return probability as a function of time.

### 4.1 Evaluation of the LDOS correlation function

Combining Eqs. (13) and (20) we find

$$\beta_+(i\varkappa, \epsilon) = \frac{1}{2\pi P_D(\epsilon)} \int\limits_{0}^{\infty} dr \frac{L_{\mu/2}(r)}{\varkappa + \gamma_0(\varkappa, \epsilon)r}. \quad (32)$$

Using $\gamma_0(\varkappa \to 0, \epsilon) = \Gamma_0$ we derive $\beta_+(0, \epsilon) = \frac{\Gamma(1+2/\mu)}{2\pi P_D(\epsilon)\Gamma_0(\epsilon)}$, hence:

$$\beta(\omega \ll \Gamma_0, \epsilon) \approx \frac{\Gamma(1 + 2/\mu)}{\pi P_D(\epsilon)\Gamma_0(\epsilon)}. \quad (33)$$

To calculate the high-frequency asymptotic of the $\beta_+(i\varkappa, \epsilon)$ make use of the Mellin transform:

$$\beta_+(i\varkappa, \epsilon) = \frac{1}{2\pi P_D(\epsilon)} \int\limits_{0-i\infty}^{0+i\infty} \frac{d\lambda}{2\pi i} \frac{2\Gamma(\frac{2}{\mu}(1-\lambda))}{\mu\gamma_0(\varkappa,\epsilon)} \Gamma(\lambda) \left(\frac{\gamma_0(\varkappa,\epsilon)}{\varkappa}\right)^\lambda \approx$$

$$\frac{1}{2\pi P_D(\epsilon)} \frac{1}{\varkappa} \left(1 - \Gamma(1+\mu/2)\left(\frac{\gamma_0(\varkappa,\epsilon)}{\varkappa}\right)^{\mu/2}\right). \tag{34}$$

Using the Eq. (31), we proceed with the analytical continuation and derive LDOS correlation function at high frequency:

$$\frac{\beta(\omega \gg \Gamma_0, \epsilon)}{\beta(0,\epsilon)} = \frac{\Gamma(1+\mu/2)\Gamma(\mu/2)}{\Gamma(1+2/\mu)\Gamma(2-2/\mu)} \cos\left(\pi(1-\mu/2)\right) \left(\frac{\Gamma_0(\epsilon)}{\omega}\right)^\mu. \tag{35}$$

The similar (power-law) behaviour of the LDOS correlation function is characteristic for the problem of Anderson localisation near the mobility edge [23].

## 4.2   Return probability

The disorder-averaged return probability for the given basis state $|j\rangle$ is defined as follows:

$$R(t) = \langle |\langle j|e^{-iHt}|j\rangle|^2\rangle = \int d\epsilon_+ d\epsilon_- e^{-i(\epsilon_+-\epsilon_-)t} \langle\rho(\epsilon_+)\rho(\epsilon_-)\rangle = \int d\epsilon d\omega e^{-i\omega t}\beta(\omega,\epsilon)\langle\rho(\epsilon)\rangle^2. \tag{36}$$

Since a given basis state generally expands over the full zone, the Eq. (36) involves integration over all the band and energy dependence of the function $\Gamma_0(\epsilon)$ matters. It's convenient to 'isolate' a specific part of the spectrum (around a certain energy $\epsilon$) for a more fine-grained analysis. Typically, this procedure is implemented by the projection of the state $|j\rangle$ on a subspace of states with almost equal energy. Resorting again to a semi-classical approximation, $h_0 \ll W$, we observe that the state $|j\rangle$ overlaps substantially only with the eigenstates of energy $\epsilon \approx H_{jj}$. Introducing a cut-off $\Lambda$, we define the energy-dependent ('truncated') return probability:

$$R_\Lambda(t|\epsilon) = \langle\theta\left(\Lambda - |H_{jj} - \epsilon|\right)|\langle j|e^{-iHt}|j\rangle|^2\rangle/(2\Lambda P_D(\epsilon)). \tag{37}$$

Let us consider the energy strip of the width $\Lambda$ satisfying $\Gamma_0 \ll \Lambda \ll W$. The later inequality guarantees that the typical width inside the interval is almost constant. The former inequality ensures that there are a lot of states with different $\gamma$ in the considered strip so we can assume that this quantity is self-averaging. This observation leads us to the following (universal) result for the 'truncated' return probability (for $t \gg \Lambda^{-1}$):

$$R_\Lambda(t|\epsilon) = P_D(\epsilon) \int d\omega e^{-i\omega t}\beta(\omega,\epsilon). \tag{38}$$

This result does not depend on $\Lambda$, and we omit this index in what follows. We consider $t > 0$ and employ the inverse Laplace transform to perform the frequency integration.

In the short-time limit, $t \ll \Gamma_0^{-1}$ the integral accumulates at $\varkappa \gg \Gamma_0$. Using the high-frequency asymptotic Eq. (34), we find:

$$\int \frac{d\varkappa}{2\pi i} e^{\varkappa t}\beta_+(\varkappa,\epsilon) \approx \frac{1}{2\pi P_D(\epsilon)}\left(1 - \frac{\Gamma(1+\mu/2)\Gamma(\mu/2)}{\Gamma(2-2/\mu)\Gamma(\mu)} (t\Gamma_0)^{\mu-1}\right). \tag{39}$$

In the long-time limit, we are not able to directly evaluate the frequency integral, since it involves a function which is only defined implicitly via Eq. (30). Let us make a certain (not fully controllable) approximation: assume that the typical scale $\gamma_0$, see Eq. (20), is independent of $\varkappa$, i.e., replace $\gamma_0(\varkappa)$ by a constant $\Gamma_0$. Under this assumption, we arrive at the following 'naive' estimate for the return probability:

$$R^{(\text{naive})}(t) = \int \frac{d\varkappa}{2\pi i} e^{\varkappa t} \int\limits_0^\infty dr \frac{L_{\mu/2}(r)}{\varkappa + \Gamma_0 r} = \int\limits_0^\infty e^{-\Gamma_0 t r} L_{\mu/2}(r) dr = \exp\{-(\Gamma_0 t)^{\mu/2}\}. \tag{40}$$

To evaluate the last integral, we used the property of the one-sided stable distribution (see Eq. (64)). This analysis results in a stretched-exponential behaviour of the return probability, with an exponent of $\mu/2$. The asymptotic behaviour in the limit $t \to \infty$ arises from the distribution $P(\gamma)$ for sufficiently small $\gamma$, which has the following asymptotic form:

$$\ln P(\gamma) \sim -(\Gamma_0/\gamma)^{\mu/(2-\mu)}. \tag{41}$$

The integral in Eq. (38) can be more precisely evaluated numerically (see the 'theory' curve in the Fig. 3). In the long-time limit, the asymptotic behaviour is given by:

$$\ln\left(R(\Gamma_0 t \gg 1)\right) \approx -C(\mu)(\Gamma_0 t)^{\mu/2}, \tag{42}$$

which exhibits the same stretched-exponential behaviour as the 'naive' estimate above, but with a different numerical coefficient.

## 5 Numerics

In this Section, we explore the properties of the Hamiltonian Eq. (1) numerically, assuming the following distributions of diagonal and off-diagonal terms:

$$P_D(\xi) = \frac{1}{\sqrt{2\pi}} \exp\left\{-\frac{\xi^2}{2}\right\} \quad P_L(h) = \frac{h_0^\mu \mu}{2N|h|^{\mu+1}} \theta(|h| - |h_0|N^{-1/\mu}), \tag{43}$$

where $\theta$ stays for the Heaviside theta function. As we have demonstrated, the only relevant frequency scale in the problem is $\Gamma_0$, see Eq. (29), i.e. after proper re-scaling all results become universal. Our first goal will be to properly identify this scale.

### 5.1 The typical scale of the decay rate

Recall that $\Gamma_0$ is defined as a scale of the distribution of the level widths, Eq. (20) for $\varkappa \to 0$. In this limit we have $\gamma(\varkappa \to 0, \epsilon) = 2\,\text{Im}(\Sigma(\epsilon))$. Using properties of the one-sided stable distribution one can find:

$$\langle \ln(\text{Im}(\Sigma)) \rangle = \ln\left(\frac{\Gamma_0}{2}\right) - \left(1 - \frac{2}{\mu}\right) \gamma_{Euler} \quad \gamma_{Euler} \approx 0.577. \tag{44}$$

This formula allows to extract $\Gamma_0$ from the distribution of $\text{Im}(\Sigma)$. In order to compute this distribution numerically, we solve the cavity equation Eq. (6) via population dynamics, also

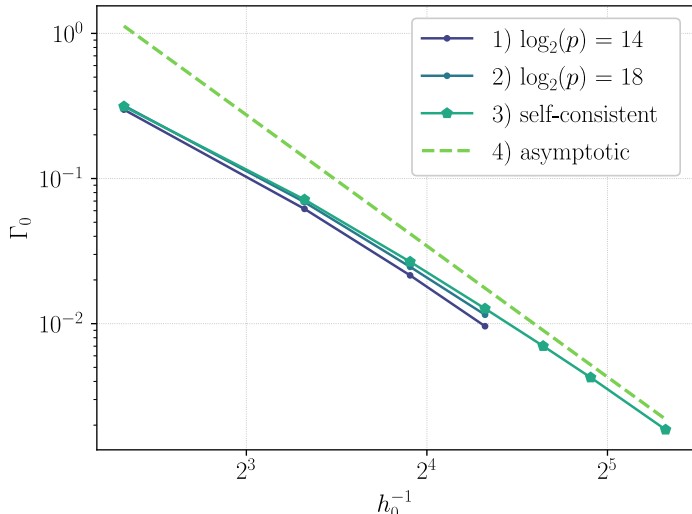

Figure 1: Comparison of $\Gamma_0$ obtained by different way: 1), 2) – solution of the cavity equation by population dynamic with different pool sizes. 3) solution of a self-consistency equation, Eq. (23) and 4) asymptotic ($h_0 \ll W$) solution of the self-consistency equation, given in Eq. (29).

known as the pool method. To this end, we consider a 'pool' of the size $p$ of Green function values $G_{jj}^{(t)}(\epsilon + i\eta)$ (here $j \in 1, \ldots, p$) and the index $t$ indicates steps of the following iteration:

$$\frac{1}{G_{jj}^{(t+1)}(z)} = z - H_{jj} - \sum_{l=1}^{p} |H_{jl}|^2 G_{ll}^{(t)}(z). \tag{45}$$

The steady state of this dynamics provides the distribution of $G$ for our matrix ensemble.

The results for $\Gamma_0$ derived with this method are plotted on the Fig. 1 for two pool sizes $p = 2^{14}$ and $p = 2^{18}$ (the lines 1 and 2, accordingly). The smaller $h_0$, the more relevant finite-pool-size effects become and reaching a large pool size limit becomes prohibitively time-consuming (the pool method requires $\mathcal{O}(p^2)$ steps, since the Lévy matrix is dense). In order to overcome this limitation, we consider an alternative method, which is based on solving the Eqs. (23), (24) directly for the relevant scales $\gamma_0$ and $R_0$. In doing so, we use the fact that for small $h_0$, the distribution factorizes: $P(\gamma, R) = P_\gamma(\gamma) P_R(R)$ (see Appendix C) and use the explicit expressions for these distributions in Eqs. (20) and (22). The result of this computation is shown on the Fig. 1, as a 'self-consistent' line. Note the excellent agreement with solution of the full distributional equation, obtained via the pool method. This 'self-consistent' approach is much more computationally efficient and allows us to obtain the relevant energy scale for much smaller values of $h_0$. One can observe that with the increase of $p$ the value of $\Gamma_0$, extracted from the pool method, asymptotically approaches the one obtained from the self-consistency equations Eqs. (23), (24), which, in turn, tends to the analytical result, Eq. (29) at the smallest $h_0$. In what follows, we will use the 'self-consistent' value as the reference one for rescaling of the results to their universal form.

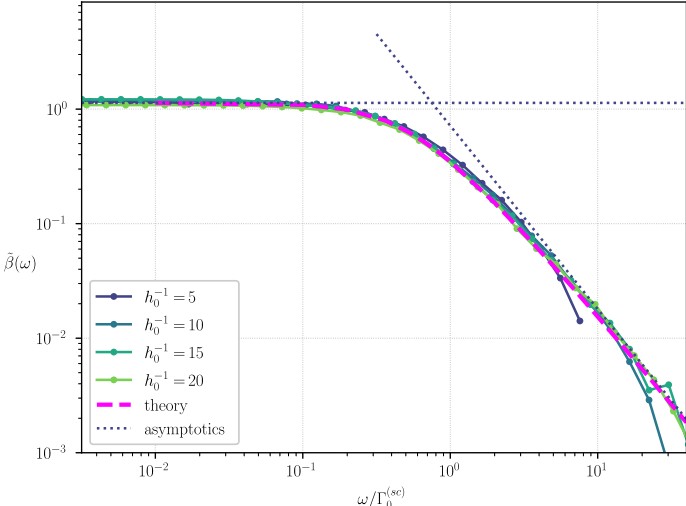

Figure 2: Re-scaled LDOS correlation function $\tilde{\beta}(\omega) = \pi P_D(\epsilon)\Gamma_0\beta(\omega)$ as a function of $\omega/\Gamma_0$ for different $h_0$ and $\mu = 1.6$. The theoretical curve is obtained as a numerical evaluation of (32). The asymptotic lines are given by Eq. (33) and Eq. (35).

## 5.2 LDOS correlation function

After having the energy scale $\Gamma_0$ evaluated numerically, we switch to the LDOS correlation function, defined in Eq. (9). In order to additionally verify applicability of the pool method, we compute this quantity from the exact diagonalization. We consider the systems of the size $N = 2^{15}$. In order to access the relevant limit of $N \to \infty$, we broaden the levels into Lorentzians of the width $\delta$ chosen in such a way such that $\Gamma_0 \gg \delta \gg N^{-1}$. The result is shown on the Fig. (2) (for $\delta = 10^{-3}$). The graph comprises the results for several values of $h_0$, with frequency rescaled to the relevant $h_0$-dependent energy scale. The results can be directly compared to the pink dashed line, which was obtained by solving numerically Eq. (32). The data collapses nicely to a universal curve, which shows complete agreement with the expected result (observe also the asymptotics, as given in Eq. (33) and Eq. (35)).

## 5.3 Return probability

As we have already discussed, the return probability is, essentially, the Fourier-transformed LDOS correlation function, see Eq. (38). Thus, in this section, we compute $R(t)$ in two ways: i) directly extracting it from the quantum dynamics, solving the Schroedinger equation, and ii) Fourier-transforming numerically the solution to Eq. (32). We consider the fixed value $\mu = 1.4$ and several values of $h_0$. For the first approach, we consider the finite systems of the size $N = 2^{15} - 2^{17}$. The results are plotted on the Fig. 3.

The reason for deviation of the numerical curves (from blue to green) from the theoretical one (dashed pink) is twofold. First, for short times $Wt \sim 1$ (equivalently, $\Gamma t \sim \Gamma/W$) the curvature of the spectrum, which is not accounted for in the theoretical computation, is not negligible. This effect becomes less severe at fixed $t\Gamma_0$ with decrease of $h_0$, as $\Gamma_0$ is decreasing. Second, for long times, the system starts to 'feel' its finite size, where the return probability saturates (the numerical curves, computed at finite system sizes, bend down at large times).

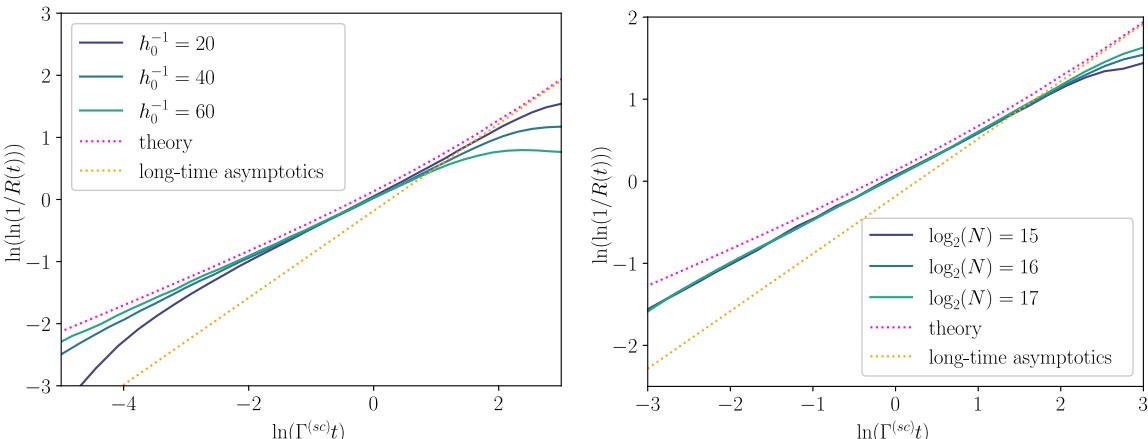

Figure 3: Return probability $R(t)$. The 'theory' curve is a numerical evaluation of the integral in Eq. (38). The asymptotic curve shows the theoretical asymptotic of return probability at sufficiently large time: $\ln(1/R(t)) \approx (0.77\,\Gamma_0 t)^{0.7}$. Left: dependence on $h_0$ at the fixed system size $N = 2^{16}$. Right: system-size dependence at fixed $h_0 = 1/20$.

The saturation value $R(t \to \infty)$ gradually decreases with increase of $N$ (see the Fig. 3, right), as expected.

# 6 Discussion

In this Section we would like to discuss the main features of the LRP model less formally. Most importantly, the FGR is not applicable due to the long-tailed distribution of the off-diagonal matrix elements, Eq. (3). One feature of the model, closely related to failure of the FGR, is a strong frequency-dependence of the typical level width, see Eq. (30). The typical level width grows with $\varkappa$ increasing. These features have been first pointed out in the Ref. [18]. However, the analysis in the Ref. [18] was based on the Wigner-Weisskopf approximation, which is valid only for $\varkappa \gg \Gamma_0$ or, equivalently, in the short-time limit of the return probability.

   This analysis in spirit of Refs. [18, 19] can be used to qualitatively understand our result at high frequency. In this regime, a perturbation theory can be used to estimate the decay rate, but with certain care. In particular, one has to relax the energy conservation condition, imposed by the $\delta$ function in the FGR, to the condition $\epsilon - \xi_l \sim \varkappa \sim \omega$. We thus write the following estimate for the typical decay rate as a function of $\omega$:

$$\gamma_0(\omega) \sim \frac{1}{\omega} \sum_l |H_{jl}|^2 \theta(\omega - |\epsilon - \xi_l|). \tag{46}$$

The sum in this equation is not self-averaging, since the expected value of $|H_{jl}|^2$ is infinite. This implies that only $O(1)$ largest terms out of $N_\omega \sim N\omega P_D(\epsilon)$ terms typically contribute. We thus have to estimate the largest element in the sum in Eq. (46). Let us introduce $w = |H_{jl}|^2$ which is distributed (at large $w$) according to $P(w) = \frac{\mu H_0^\mu}{2Nw^{\mu/2+1}}$, see Eq. (3). The cumulative density of $w$ at large argument is $F(w) = 1 - \int_w^\infty P(w')dw'$, and the cumulative

density of the largest of $N_\omega$ terms reads:

$$F_M(w) = F(w)^{N_\omega} = \exp\left\{-\frac{N_\omega}{N}\frac{h_0^\mu}{w^{\mu/2}}\right\}. \tag{47}$$

This estimation shows that the largest term is of the order $w_{max} \sim h_0^2(\omega P_D(\epsilon))^{2/\mu}$, and the the typical level width becomes:

$$\gamma_0(\omega) \sim w_{max}/\omega \sim \frac{h_0^2}{\omega}(\omega P_D(\epsilon))^{2/\mu}. \tag{48}$$

The value of $\Gamma_0 = \gamma_0(\omega \to 0)$ can be determined self-consistently from the condition $\omega \sim \Gamma_0 \sim \gamma_0$ i.e. $\Gamma_0^{\mu-1} \sim h_0^\mu P_D(\epsilon)$. One can see that these qualitative estimations coincide with Eqs. (29), (31).

The correlation function of LDOS $\beta(\omega)$ can be estimated using Eq. (13). Observe that the main contribution to $\beta(\omega)$ comes from the region $\gamma \sim \omega$. The probability to find the width of this order of magnitude is $(\gamma_0(\omega)/\omega)^{\mu/2}$ which leads to the following estimate:

$$\beta(\omega) \sim \frac{1}{P_D(\epsilon)}\frac{1}{\omega}(\gamma_0(\omega)/\omega)^{\mu/2} \sim \frac{1}{P_D(\epsilon)\Gamma_0}(\Gamma_0/\omega)^{-\mu}. \tag{49}$$

The above analysis correctly describes asymptotic behaviour of the LRP model, however for the correct prediction of the long-time behaviour of the return probability it is required to know the full behaviour of the LDOS correlation function (see Fig. 3).

In discussing the numerical results, we would like to point out that $\Gamma_0$ converges slowly to its asymptotic behaviour Eq. (29), see also the Fig. 1. For example, one needs $h_0/W \sim 1/50$ (and $\Gamma_0/W \sim 10^{-3}$) to observe the limiting scaling law of $\Gamma_0$ for $\mu = 1.5$. This observation can explain the difference between the theoretical prediction and the numerical result of the Ref. [19] (see Fig. 4 of this reference). Another manifestation of the slow convergence to the $h_0 \to 0$ limit would show up in the attempt to observe the properties of the non-ergodic extended phase. Recall that this phase would be observed under the following scaling of the power-law tail with the system size: $h_0^\mu = N^{1-\eta}$ for $\eta < \mu$. In order to reach the ultimate $h_0 \to 0$ asymptotic behaviour, it would take the systems of the size $N > (1/h_0)^{\mu/(\mu-1)}$ which i. e. for $\mu = 1.5$ becomes $N > (50)^3 \approx 2^{17}$ (using the above estimate for the required $h_0$).

Finally, we would like to put our results into the context of the MBL studies. Even though we are not aware of a direct mapping between the LRP ensemble and MBL problems, we find the Ref. [15] illuminating in this regard. The authors performed the Jacobi algorithm for the many-body Hamiltonian $H$. This algorithm repeatedly performs rotations in the two-dimensional subspace $(|i\rangle, |j\rangle)$ until the matrix becomes almost diagonal. The subspace for rotations is chosen at each step such that $H_{ij}$ is the largest off-diagonal element of the current $H$. The key observation of the Ref. [15] is that the distribution function of the selected $H_{ij}$ has a power-law tail $P(H_{ij}) \sim |H_{ij}|^{-1-\mu}$, where $\mu$ depends on details of the many-body problem, such as the interaction strength and the degree of disorder. The same tail characterizes the distribution of decimated elements in the application of the Jacobi algorithm to LRP matrices. In this case, the index $\mu$ coincides with the index from the distribution in Eq. (2).

This similarity suggests comparing the results for certain quantities between the two models. For example, the authors of Ref. [15] compute the spin-spin correlation function and relate the stretch exponent of its decay to the exponent of the power-law decay of the distribution of the matrix elements, decimated in the Jacobi algorithm, suggesting that the stretch exponent

should be $\mu - 1$. Our computation for the similar quantity, namely the return probability in the LRP model, results into a different relation of the stretch exponent to the exponent of the power-law tail, $\mu/2$. Even though the difference can be related to finer details of the two models, our computation allows to anticipate the magnitude of the finite-time corrections to the true asymptotic behaviour. As the Fig. 3 illustrates, the return probability as a function of time *looks* like a straight line up to relatively long times $\ln(\Gamma_0 t) \approx 1.5$. The slope of this line, which is actually the short-time asymptotic, see Eq. (39), is indeed $\mu - 1$. This short-time asymptotic, if misinterpreted, may hinder extraction of the true (long-time limit) stretch exponent for the LRP model and, possibly, in the actual MBL problem, too.

## Acknowledgements

A.V.L. would like to thank M.V. Feigel'man and V.E. Kravtsov for fruitful and enlightening discussions.

## A  Generalised central limit theorem and one-sided stable distribution

### A.1  Laplace transform of the tailed distribution

Consider a positive-valued random variable which has a distribution with the power-law asymptotic:

$$P(x \to \infty) \approx \frac{\alpha}{x^{\nu+1}} \quad \nu < 1 \tag{50}$$

We will be, in particular interested in the distribution in Eq. (3), that is in our case: $\nu = \mu/2$ and $\alpha = \mu h_0^\mu/(2N)$. We are interested in computing the Laplace transform:

$$\hat{P}(s) = \int_0^\infty dx e^{-sx} P(x) = \int_{+0-i\infty}^{+0+i\infty} s^{-\lambda} \Gamma(\lambda) \mathcal{M}(P)(1-\lambda) \frac{d\lambda}{2\pi i} \tag{51}$$

Here $\mathcal{M}$ is the Mellin transform:

$$\mathcal{M}(P)(1-\lambda) = \int_0^\infty x^{-\lambda} P(x) dx. \tag{52}$$

Note that this function has a pole at $\lambda = -\nu$:

$$\mathcal{M}(P)(1-\lambda) \underset{\lambda \to -\nu+0}{\approx} \int_\Lambda^\infty x^{-\lambda} \frac{\alpha}{x^{\nu+1}} dx \approx \frac{\alpha}{\lambda + \nu} \tag{53}$$

and there are no other poles in the strip $-\mu < \operatorname{Re}\lambda \le \epsilon$ for some small $\epsilon > 0$. Finally, we can evaluate the integral in Eq. (51) by summing over the residues located in the left half-plane. As we are only interested in the behaviour of this function at small arguments, we only need two poles: $\lambda = 0, -\nu$. This brings us to the result:

$$\hat{P}(s) \approx 1 + \alpha \, \Gamma(-\nu) s^\nu \tag{54}$$

## A.2 Generalised central limit theorem

In this section we are interested in the distribution of the sum of $N$ i.i.d. random variables drawn from the heavy-tailed distribution, Eq. (50):

$$X = \sum_{j=1}^{N} x_j. \tag{55}$$

The probability density function of $X$ is given by the inverse Laplace transform:

$$P(X) = \int_{-i\infty}^{i\infty} \frac{ds}{2\pi i} \langle \exp\{sX - s\sum_j x_j\}\rangle. \tag{56}$$

To take the average over $x_j$, we notice that $x_j s \sim x_j/X \ll 1$, so we can use the approximate expression for the Laplace transform in Eq. (54):

$$P(X) = \int_{-i\infty}^{i\infty} \frac{ds}{2\pi i} \langle \exp\{sX + N\alpha\,\Gamma(-\nu)s^\nu\}\rangle. \tag{57}$$

Note that the natural scaling for $\alpha$ to have a proper limit $N \to \infty$ is $\alpha \sim 1/N$. Introducing $X_0 := [-N\alpha\Gamma(-\nu)]^{1/\nu}$ we conclude that the random variable $X/X_0$ has the probability density function

$$P(X/X_0) = \int_{-i\infty}^{i\infty} \frac{ds}{2\pi i} \exp\{sX/X_0 - s^\nu\}. \tag{58}$$

This is exactly the one-sided stable distribution.

## A.3 One-sided stable distribution

The probability density of the one-sided stable distribution is determined by the following inverse Laplace transform:

$$L_\nu(\zeta) = \int_{-i\infty}^{i\infty} \frac{ds}{2\pi i} \exp\{s\zeta - s^\nu\}. \tag{59}$$

The asymptotic at large arguments is:

$$L_\nu(\zeta \gg 1) \approx \int_{-i\infty}^{i\infty} \frac{ds}{2\pi i} \exp\{s\zeta\}(-s^\nu) = \frac{\nu\,\zeta^{-\nu-1}}{\Gamma(1-\nu)}. \tag{60}$$

The asymptotic at the small argument can be calculated using the saddle-point approximation:

$$L_\nu(\zeta \ll 1) \approx \frac{1}{\sqrt{2\pi\nu(1-\nu)\left(\frac{\zeta}{\nu}\right)^{(2-\nu)/(1-\nu)}}} \exp\left\{-(1-\nu)(\nu/\zeta)^{\nu/(1-\nu)}\right\}. \tag{61}$$

The final part is the Mellin transform of the stable distribution:

$$\mathcal{M}(L_\nu)(\lambda) = \int_0^\infty \zeta^{\lambda-1} L_\nu(\zeta) d\zeta. \tag{62}$$

This integral is defined for $\operatorname{Re}\lambda < 1+\nu$ and the result is an analytic function of $\lambda$. We assume $0 < \lambda < 1$ and then perform the analytical continuation. We will need the following auxiliary integral:

$$\zeta^{\lambda-1} = \frac{1}{\Gamma(1-\lambda)} \int_0^\infty r^{-\lambda} e^{-r\zeta} dr. \tag{63}$$

Using the integral

$$\int_0^\infty e^{-s\zeta} L_\nu(\zeta) d\zeta = e^{-s^\nu}, \tag{64}$$

we derive the result (recall $\operatorname{Re}\lambda < 1+\nu$):

$$\mathcal{M}(L_\nu)(\lambda) = \int_0^\infty \frac{d\zeta}{\Gamma(1-\lambda)} \int_0^\infty r^{-\lambda} e^{-r\zeta} dr L_\nu(\zeta) =$$
$$\frac{1}{\Gamma(1-\lambda)} \int_0^\infty r^{-\lambda} e^{-r^\nu} dr = \frac{\Gamma((1-\lambda)/\nu)}{\nu\Gamma(1-\lambda)}. \tag{65}$$

# B  Applicability of the cavity equation

We aim to provide an estimate that supports the validity of the cavity equation for our problem. We will follow the steps, outlined in the Ref. [16], specifically following the discussion below Eq. (6) of the mentioned reference.

In the main text the cavity equation was written for the diagonal part of the Green function, Eq. (6). Let us now write an equation including the off-diagonal parts:

$$\begin{pmatrix} G_{00}(z) & G_{01}(z) \\ G_{10}(z) & G_{11}(z) \end{pmatrix}^{-1} = \begin{pmatrix} z-H_{00} & -H_{01} \\ -H_{10} & z-H_{11} \end{pmatrix} - \begin{pmatrix} \Sigma_{00}(z) & \Sigma_{01}(z) \\ \Sigma_{10}(z) & \Sigma_{11}(z) \end{pmatrix} \quad \Sigma_{ab}(z) = \sum_{l,m\neq 0,1} H_{al} G_{lm}^{\widehat{01}}(z) H_{mb}, \tag{66}$$

where $G_{lm}^{\widehat{01}}(z)$ is a Green function of the matrix obtained from the matrix $H$ (given by the Eq. (1)) by excluding the 0th and 1st rows and columns. We assume the smallness of the off-diagonal terms and check self-consistency of this assumption. Under this assumption, the following approximate relation can be written:

$$G_{01}(z) \approx G_{00}(z)(\Sigma_{01}(z) + H_{01})G_{11}(z) \quad \Sigma_{01}(z) = \Sigma_{01}^{(d)}(z) + \Sigma_{01}^{(o)}(z)$$
$$\Sigma_{01}^{(d)}(z) = \sum_{l\neq 0,1} H_{0l} G_{ll}^{\widehat{01}}(z) H_{l1} \quad \Sigma_{01}^{(o)}(z) = \sum_{l,m\neq 0,1; l\neq m} H_{0l} G_{ml}^{\widehat{01}}(z) H_{m1} \tag{67}$$

The quantities $\Sigma_{01}(z)^{(d,o)}$ are random variables. To estimate their typical values, we apply the generalized central limit theorem twice:

$$|\Sigma_{01}(z)^{(d)}|_{typ} \sim h_0 \left[\frac{1}{N}\sum_{l\neq 0,1} |G_{ll}^{\widehat{01}}(z) H_{l1}|^\mu\right]^{1/\mu} \sim \left[\frac{h_0^{2\mu}}{N}\langle |G_{ll}^{\widehat{01}}(z)|^\mu\rangle\right]^{1/\mu}$$

$$|\Sigma_{01}(z)^{(o)}|_{typ} \sim \left[\frac{h_0^\mu}{N}\sum_l \left(\sum_m |G_{ml}^{\widehat{01}}|(z)|H_{m1}|\right)^\mu\right]^{1/\mu} \sim \left[h_0^{2\mu}\langle |G_{ml}^{\widehat{01}}(z)|^\mu\rangle\right]^{1/\mu} \tag{68}$$

The $\langle |G_{ll}|^\mu\rangle$ remains finite due to the non-zero level width. From the above estimation, we observe that $G_{01} \sim \Sigma_{01}(z) \sim N^{-1/\mu}$. Therefore, we can conclude that the cavity equation holds in our case.

## C  Joint probability distribution of real and imaginary parts of self-energy

In this Appendix we evaluate the joint probability distribution of real and imaginary parts of the self-energy, defined in Eqs. (14) and (21). We also argue that these random variables are independent in the limit of $W \gg h_0$. To begin, we notice that the joint distribution of $R$ and $\gamma$, with the use of the cavity equation Eq. (6), can be written as follows:

$$P(R, \gamma | \varkappa, \epsilon) = \int\limits_{-\infty}^{\infty} \frac{dt}{2\pi} \int\limits_{-i\infty}^{i\infty} \frac{d\tau}{2\pi i} \exp\{itR + \gamma\tau$$

$$- \sum_{l \neq j} |H_{jl}|^2 \left[ \left( \frac{it}{2} + i\tau \right) G_{ll}^{\hat{j}} \left( \epsilon + \frac{i\varkappa}{2} \right) + \left( \frac{it}{2} - i\tau \right) G_{ll}^{\hat{j}} \left( \epsilon - \frac{i\varkappa}{2} \right) \right]\}. \tag{69}$$

This expression can be averaged over $H_{jl}$ to give:

$$P(R, \gamma | \varkappa, \epsilon) = \int\limits_{-\infty}^{\infty} \frac{dt}{2\pi} \int\limits_{-i\infty}^{i\infty} \frac{d\tau}{2\pi i} \exp\{itR + \gamma\tau - F(t, \tau)\}, \tag{70}$$

where

$$F(t, \tau) = \frac{h_0^\mu}{N} \Gamma\left( 1 - \frac{\mu}{2} \right) \sum_{l \neq j} \left[ \left( \frac{it}{2} + i\tau \right) G_{ll}^{\hat{j}} \left( \epsilon + \frac{i\varkappa}{2} \right) + \left( \frac{it}{2} - i\tau \right) G_{ll}^{\hat{j}} \left( \epsilon - \frac{i\varkappa}{2} \right) \right]^{\mu/2}. \tag{71}$$

The quantity $F$ is self-averaging, similarly to the 'action' in Eq. (18). In a more explicit form, it reads:

$$F(t, \tau) = h_0^\mu \Gamma\left( 1 - \frac{\mu}{2} \right) \int P_D(\xi) d\xi \int dR d\gamma P(R, \gamma | \varkappa, \epsilon) \left[ \frac{it(\epsilon - R - \xi) + \tau(\varkappa + \gamma)}{(\epsilon - R - \xi)^2 + \left( \frac{\varkappa + \gamma}{2} \right)^2} \right]^{\mu/2} \tag{72}$$

Let us first consider the probability distribution of $R$:

$$P(R | \varkappa, \epsilon) = \int P(R, \gamma | \varkappa, \epsilon) d\gamma = \int\limits_{-\infty}^{\infty} \frac{dt}{2\pi} \exp\{itR - F(t, 0)\}. \tag{73}$$

This is, in fact, a stable Lévy distribution, which in the standard form becomes:

$$P(R | \varkappa, \epsilon) = \frac{1}{R_0(\varkappa, \epsilon)} L_{\mu/2}^{\beta(\varkappa, \epsilon)}(R/R_0(\varkappa, \epsilon)) = \int\limits_{-\infty}^{\infty} \frac{dt}{2\pi} \exp\{itR - |t|^{\mu/2} R_0(\varkappa, \epsilon)^{\mu/2} e^{-i \operatorname{sgn}(t)\beta_0(\varkappa, \epsilon)}\}. \tag{74}$$

The expression for $R_0(\varkappa, \epsilon)$ is given in the main text, see Eq. (24). In the limit $h_0 \ll W$ we can derive a simplified expression, starting directly from Eq. (72). Indeed, the integral for $F(t, 0)$ is dominated by $\xi$, satisfying $\epsilon - \xi \sim W$, while typical $R$ and $\gamma$ are much smaller then $W$ and can be neglected. This gives:

$$P(R | \epsilon) = \int\limits_{-\infty}^{\infty} \frac{dt}{2\pi} \exp\{itR - h_0^\mu |t|^{\mu/2} \Gamma\left( 1 - \frac{\mu}{2} \right) \int P_D(\xi) d\xi \frac{1}{|\epsilon - \xi|^{\mu/2}} e^{i \operatorname{sgn}(t(\epsilon - \xi))\pi\mu/4}\}, \tag{75}$$

or simply $F(t,\tau) \sim (t/W)^{\mu/2} h_0^\mu$. Substituting this into Eq. (73) we find that the typical value of $R$ is of the order $h_0^2/W$ and the integral in this expression is coming from $t \sim W/h_0^2$ (see also a related observation in Ref. [22]).

Similar analysis can be done for distribution of $\gamma$. In this case, we need to compute the integral for $F(0,\tau)$. This estimation is more involved: one has to take into account that the integral over $\gamma$ is coming from $\gamma \sim \Gamma_0$ (see Eq. (23)) and as a result the $\xi$-integration is dominated by the region $\epsilon - R - \xi \sim \Gamma_0$. Finally, $F(\tau) \sim (\tau\Gamma_0)^{\mu/2}$ and $\tau$ contributing to the integral in Eq. (19) can be estimated as $\tau \sim \Gamma_0^{-1}$.

The two observations above can be combined to estimate $F(t,\tau)$ in the relevant domain $\tau \sim \Gamma_0^{-1}$ and $t \sim W/h_0^2$. The integration over $\xi$ can be separated into two regions. In the first region, $\epsilon - R - \xi \sim W$, one has $(\epsilon - R - \xi)t \sim (W/h_0)^2$ which is much larger then $\tau(\varkappa + \gamma) \sim 1$, hence, one can put $\tau \to 0$ while evaluating the contribution of this region to the integral. In the second region, $\epsilon - R - \xi \sim \Gamma_0$, one has $(\epsilon - R_j - \xi)t \sim \Gamma_0$ which is much larger then $\tau(\varkappa + \gamma_j) \sim 1$ and hence one can put $t \to 0$. As a result:

$$F(t,\tau) \approx F(t,0) + F(0,\tau), \tag{76}$$

which, substituted into Eq. (70) leads to $P(R,\gamma|\varkappa,\epsilon) \approx P(R|\epsilon)P(\gamma|\varkappa,\epsilon)$.

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
