# Peer review of "Local Density of States Correlations in the Lévy-Rosenzweig-Porter random matrix ensemble"

_SciPost Physics_

## Round 1 · Referee Report · Carlo Vanoni (Referee 1) · 2024-12-30

Strengths

1- The results presented in the paper are solid 2- The analytical calculations are presented with enough details 3- The analytical results are verified numerically and they agree

Weaknesses

1- There are some minor mistakes in two equations 2- There are some typos in the text 3- A comment in the conclusions of the paper can be improved

Report

This paper presents an analytical calculation of the finite-frequency correlation function of the local density of states of Lévy-Rosenzweig-Porter random matrices, using a "semi-classical" approximation in the computation of the imaginary part of the self-energy. The results presented in the paper are compared with numerical results and the authors find good agreement with the analytical predictions in the desired range of validity of their approximations. Despite being valid for this specific class of random matrices and probably not directly applicable to more complicated models, I find the results interesting and worthy of publication in SciPost Physics.

I report here some comments about the manuscript.

1- Eqs. (4), (5) and (6) are inconsistent, there must be some signs that are not correct. I believe that the definition of self-energy reported as the second part of Eq. (4) is wrong, and should be $\Sigma_j(z) = z - H_{jj}- 1/G_{jj}(z)$. This way the usual textbook definition is met and the rest of the paper is consistent with it. 2- I think that a factor 2 is missing when going from Eq. (11) to Eq. (12) when the authors define an auxiliary imaginary frequency. I think the right definition should be $ix = \omega + 2 i \delta$. Or instead, $ix = \omega/2 + i \delta$, removing the factor of 2 in ($ix / 2$) in the following equations. I think this is not affecting the validity of the results, just a matter of consistency in the definitions. 3- In the conclusions, the authors discuss their results with the findings of a previous paper discussing the Jacobi algorithm applied to MBL. The connection they find is very interesting, but there are a couple of questions I would like to ask. According to Fig. 3, the authors find that $\mu/2 = 0.7$, so that $\mu = 1.4$, while the other paper argues that the same curve should have slope $\mu - 1 = 0.4$. The difference between 0.7 and 0.4 is not small, but not even huge, and might correspond to the behavior before the long-time asymptote. Could the authors add a curve to Fig. 3 corresponding to the slope $\mu - 1$ and verify if it is consistent with this observation? Also, in the reference the authors compare to, the behavior with slope $\mu - 1$ is not always valid, but according to Eq. (15) in that reference, that behavior is valid in an intermediate time regime. Could the authors comment more on this and check if their results are valid in the same regime?

Requested changes

Some minor changes to the text.

1- I would remove "the" in "the section #" at the end of the Introduction, it doesn't sound correct. Also, at the very beginning of the Introduction "The Anderson localization" -> "Anderson localization". 2- In the first paragraph of the Introduction "neglect or simplification" -> "neglect or simplify" 3- Below Eq. (3), "...with the result is..." is not correct, maybe use "...with the result being..."? 4- Below Eq. (8), "This means that the integral is coming from $\xi \approx \epsilon$". Do the authors mean that the main contribution to the integral comes from $\xi = \epsilon$ and thus the result can be obtained by replacing the integral with the value at $\epsilon$? 5- Below Eq.(20) "The similar equation" -> "A similar equation". 6- Below Eq.(36) it is not clear what "expands over the full zone" means. Can the authors be more clear? 7- In Eq. (37) there is an unmatched $\langle \rangle$ pair. 8- In Sec. 5.2, "chosen in such a way such that" -> "chosen in a way such that".

Recommendation

Publish (easily meets expectations and criteria for this Journal; among top 50%)

---

## Round 1 · Referee Report · Anton Kutlin (Referee 2) · 2025-1-21

Strengths

1 - The paper addresses an important and timely topic, investigating LDOS correlations in the Lévy-Rosenzweig-Porter ensemble, a model of growing interest in statistical physics and random matrix theory.
2 - The results appear plausible and could contribute to understanding the dynamics of non-ergodic phases.
3 - The analytical approach allows for studying large-finite-size effects in the Levy-RP model, which has never been done before and could potentially stimulate further research.

Weaknesses

1 - The analytical derivations could benefit from additional rigor, with more detailed explanations of intermediate steps and conditions for approximations. This would enhance the clarity and verifiability of the results.
2 - The numerical experiments are interesting but lack details on parameter choices and implications. Providing these details and expanding the scope of the experiments to include measurements explicitly within the fractal phase would strengthen the analysis.
3 - The paper presents results that differ from prior work but does not sufficiently discuss the reasons for these differences. Including such a discussion would help readers better understand the novelty and implications of the findings.
4 - While the paper addresses an interesting problem, it would be improved by adding references to relevant prior studies and situating its contributions within the broader context of the field.
5 - The presentation, including notation and terminology, could be made more consistent and accessible to a general audience. Clarifying definitions and avoiding jargon would significantly enhance readability and impact.

Report

The paper presents an analytical and numerical investigation of the local density of states (LDOS) correlations in the Lévy-Rosenzweig-Porter (LRP) random matrix ensemble. The approach seems reasonable, and the results seem plausible; however, some of the claims differ from the previously known results, and the paper would benefit from discussing these discrepancies and adding more details on the analytical derivations and numerical experiments. Below, I provide my questions to the authors and suggestions for improvement.

  1. Could you please expand the discussion about non-commuting limits between Eqs. 7 and 8 and compare it to the approach chosen in, e.g., [1-4]? As you can see from these (missing) references, the studies of fractal phases in Rosenzweig-Porter models usually exploit simultaneous and not consecutive limits with a specific scaling relation between $N$ and $\delta,$ $\delta\gg W/N$. This approach has its limitations, e.g., it does not allow accessing localized phase, so it would be exciting to know if your approach has the same limitations or not. If it does, it will imply the lower bound on $h_0$ for fixed $\mu$, $h_{AT}(\mu)$, corresponding to the Anderson localization transition. 1.1 There is also a slight inconsistency of the notation: the imaginary part of the energy, i.e., the artificial broadening, is denoted by $\delta$ in most of the paper, but also by $\eta$, e.g., before Eq. 45. Please either explain the reason for the difference or make the notation consistent throughout the paper. In addition, notice that $\eta$ has been used in Eq. 2 for a different role, which might confuse readers. Using a distinct symbol here could enhance clarity.

  2. The absence of correlations between the real and imaginary parts of the self-energy, as claimed on p.10 after Eq. 45 and derived in Appendix C, appears to differ from the findings in [4], where such correlations were shown to contribute to measurable effects (see the discussion around Fig. 8 and the Appendices of [4]). This may indicate that you work in a different limit or regime where these correlations become negligible. To clarify this point, it would be valuable to test the approximation of independent real and imaginary parts of the self-energy through, e.g., numerical sampling of their distributions or analytical analysis of the corresponding approximations applicability (or both).

  3. What would strengthen the paper even more is providing more details on the specific parameters the numerical experiments were performed with and analyzing the reasons for choosing such parameters and the implications of these choices. In particular, what are the values of $\mu$ and $\delta$ in the Fig.1? Since, according to (45), the pool size p is also a system size N, my slope estimation gives $\mu=1.5$, meaning that, for $N=2^{14}$ and showed points, the system stays well enough inside the fractal phase; however, the decrease of $h_0$ for fixed $p=N$ and $\mu$ drives the system to the localized phase, raising the question of the $h_{AT}(\mu)$ importance once again. In addition, a few more comments on the self-consistent solution would be helpful: what limit does it physically correspond to? It cannot be the thermodynamic limit since this limit means $h_0=0$, and it also cannot be the $\delta\to 0$ since the asymptotic straight line supposedly describes this limit, so what values of $N$ and $\delta$ does the self-consistent solution correspond to then?

  4. Another question addressing the numerical parameters choice relates to Fig.3: calculating the corresponding values of the exponent $\eta$ introduced in (2) and (3) and analyzing these values using [5], one realizes the results from Fig.3 correspond to the Anderson transition area, with some of the measurements done above and others below the critical point but all in close proximity to it. The fact that the proposed analytical theory supposedly works at the critical point and on both sides of the transition requires additional comments. It would be desirable to have more numerical experiments inside the fractal and localized phases to demonstrate the wideness of the theory’s applicability range better. In addition, given the observable finite-size effects, a finite-size scaling analysis explicitly demonstrating the convergence to the analytically predicted thermodynamic limit would considerably strengthen the paper, especially given that the presented results differ from the results of the (missing) reference [6]; this difference and its reasons should also be explicitly discussed.

  5. Finally, I suggest adding more details on the mathematical derivations, including all intermediate steps and the derivations of the approximations validity estimations. Including these details would significantly enhance the paper's clarity for a broad audience, allowing readers to follow the derivations and reproduce the results more easily. In light of the ongoing reproducibility challenges in science, such additions, along with publishing the code used for the numerical simulations, would align well with SciPost’s commitment to Open Science and reproducibility. These steps would also help clarify the paper’s claims and strengthen confidence in its results, particularly in light of differences with some prior studies. In particular: 5.1 What is the precise mathematical meaning of the “semi-classical approximation”? In other words, what does “$\approx$” used after (8) mean specifically? How extensive is the range of $\xi$ around $\epsilon$ that contributes to the integral? How is this range related to the condition $h_0\ll W?$ See the term “Asymptotic relations,” e.g., at [7], for more details on the possible mathematical interpretations of “$\sim$” and “$\approx$.” 5.2 Given the complexity of the topic you are dealing with and to reduce the possible ambiguity, I would suggest clarifying what you mean by "self-averaging" and stating the conditions for the sum in (18) to self-average in your sense. According to [4], one can say that the resolvents’ distribution has heavy tales, and it prevents the sums of such resolvents from self-averaging, at least in the sense discussed in, e.g., [2], where it was understood as the vanishing of the ratio between the variance and the mean in the thermodynamic limit. This fact may or may not affect the paper's result, but it is better to clarify this subtle point. 5.3 What are the applicability conditions of the approximations from Appendices 1 and 2, and how do they restrict the applicability of (18)? I may guess that the applicability conditions will likely limit (20) and (22) to the tails only, which should be good enough to proceed and still get the same results, but this is better to discuss explicitly. 5.4 How does the “semi-classical approximation” help deriving (27)? What I guess you were trying to do is to substitute $\xi\to\xi+\epsilon-R$ in (23) and then exploit the fact that $P_D(x)$ is changing slowly on the scale of $\kappa+\gamma$, that’s why one can take $P_D(x)$ out from under the integration over $\xi$ replacing $P_D(\xi+\epsilon-R)$ with just $P_D(\epsilon-R)$; but in this case, one have to still deal with the integration over $R$ and make additional assumptions about the distribution of $R$ which are not explicitly specified. Anyway, I could not understand it, so adding more intermediate steps here would be helpful. 5.5 What do you mean by the “basis state” before (36)? If this is the eigenstate of the position operator, it would probably be better to state it explicitly; otherwise, it is unclear what basis is referred to. 5.6 A more detailed step-by-step derivation of this crucial result (38) would be helpful. The preceding paragraph says that the quantity $\gamma$ is assumed to be self-averaging, but Eq. (20) unequivocally says $\gamma$’s distribution has heavy tails, so why do they not spoil the self-averaging? 5.7 It seems possible to provide the applicability conditions for the “not fully controllable” approximation before (40) by analyzing Eq. 30; can it be done?

  6. Finally, I suggest clarifying the meaning of $\Gamma_0^{(sc)}$ from Fig. 2 and 3 and its relation to $\Gamma_0$, as well as putting the "new energy scale $\Gamma_0$" (the quote from the end of p.3) in the context of previous works; e.g., the scale has already been discussed in [5] as $E_{Th}$ and used later in, e.g., [4].

References [1] V.E. Kravtsov, B.L. Altshuler, and L.B. Ioffe, "Non-ergodic delocalized phase in Anderson model on Bethe lattice and regular graph," https://doi.org/10.1016/j.aop.2017.12.009

[2] A.G. Kutlin and I.M. Khaymovich, "Emergent fractal phase in energy stratified random models," https://doi.org/10.21468/SciPostPhys.11.6.101

[3] D. Facoetti, P. Vivo, and G. Biroli, "From non-ergodic eigenvectors to local resolvent statistics and back: A random matrix perspective," https://doi.org/10.1209/0295-5075/115/47003

[4] A.G. Kutlin and I.M. Khaymovich, "Anatomy of the eigenstates distribution: A quest for a genuine multifractality," https://doi.org/10.21468/SciPostPhys.16.1.008

[5] G. Biroli and M. Tarzia, "Lévy-Rosenzweig-Porter random matrix ensemble," https://doi.org/10.1103/PhysRevB.103.104205

[6] I.M. Khaymovich and V.E. Kravtsov, "Dynamical phases in a multifractal Rosenzweig-Porter model," https://doi.org/10.21468/SciPostPhys.11.2.045

[7] C. Bender, "Lectures on Mathematical Physics," https://youtu.be/LYNOGk3ZjFM?si=JAm9aveqxdtWXDIf&t=2924

Requested changes

1 - Expand the discussion of non-commuting limits in the derivations and compare the chosen approach to prior work. 2 - Provide more details on the numerical experiments, including parameter choices and their implications, and include more experiments and the finite-size scaling analysis explicitly within the fractal phase or other areas you expect your approach to work in. 3 - Address the differences between the paper’s results and prior work by discussing whether they arise from distinct limits, approximations, or methodologies. 4 - Add missing references to relevant studies and clarify the relationship between the paper’s results and prior findings. 5 - Include more intermediate steps in mathematical derivations and clarify the conditions under which approximations are valid. 6 - Address inconsistencies in notation and improve clarity in technical terms and definitions. 7* - Publish the code used for numerical simulations to enhance reproducibility and transparency; this change is "optional" because it is not yet a widespread requirement in the filed but very much desirable in light of the SciPost mission.

Recommendation

Ask for major revision

---

## Editorial Decision

resubmitted